# Evolution and Li Mineralization of the No. 134 Pegmatite in the Jiajika Rare-Metal Deposit, Western Sichuan, China: Constrains from Critical Minerals

Zhen Wang [1,2], Jiankang Li [1], Zhenyu Chen [1,*], Qinggao Yan [1,3], Xin Xiong [1], Peng Li [1] and Jingyi Deng [4]

1   MNR Key Laboratory of Metallogeny and Mineral Assessment, Institute of Mineral Resources, Chinese Academy of Geological Sciences, Beijing 100037, China; wangzhen9329@126.com (Z.W.); Li9968@126.com (J.L.); qinggaoy@pku.edu.cn (Q.Y.); XiongXin_1989@163.com (X.X.); Lipeng031111@163.com (P.L.)
2   State Key Laboratory of Mineral Processing, BGRIMM Technology Group, Beijing 102628, China
3   Laboratory of Orogenic Belts and Crust Evolution, School of Earth and Space Sciences, Peking University, Beijing 100871, China
4   School of the Earth and Land Resources, China University of Geosciences (Beijing), Beijing 100083, China; jingyideng@cugb.edu.cn
*   Correspondence: chenzhenyu@cags.ac.cn

**Abstract:** The Jiajika rare-metal deposit located in western Sichuan Province (China) is renowned as the largest lithium deposit in Asia, and the No. 134 pegmatite dike is the largest lithium pegmatite under mining conditions in the area. On the basis of a detailed characterization of textures and minerals in the Jiajika No. 134 pegmatite, two zones (the barren Zone I and the spodumene Zone II) and three subzones (Zone II was subdivided into microcrystalline, medium-fine grained and coarse-grained spodumene zones) have been identified. The detailed mineralogical characteristics of lithium minerals and other indicator minerals from each zone were evaluated by EPMA for illustrating the magmatic–hydrothermal evolution and the cooling path of the Jiajika No. 134 pegmatite. From the outer zone inwards, grain size gradually increased, the typical graphic pegmatite zone was absent, and spodumene randomly crystallized throughout nearly the whole pegmatite body. This evidence indicated a Li-saturated melt prior to pegmatite crystallization, which could be the main cause of the super-large-scale Li mineralization of the Jiajika No. 134 pegmatite. A comparison of the Cs content between primary beryl in the Jiajika No. 134 pegmatite and other important Li-Cs-Ta pegmatites in the world indicates that No. 134 pegmatite shows a high degree of fractional crystallization. The evolution of mica species from muscovite to Li-micas from Zone I to Zone II marks the transition from the magmatic to the hydrothermal stage in pegmatite evolution. The absence of individual lepidolite and the relatively limited scale of alteration of spodumene (<10 vol%) suggest that the activity of the hydrothermal fluids in the system is limited, which contributes to the preservation of the easily altered Li ores and is also an important controlling factor of the super-large-scale Li mineralization of the pegmatite. Spodumene–quartz intergrowth (SQI) usually occurs partly along the rims of the spodumene grains in the Jiajika No. 134 pegmatite. Combined with the pegmatite mineral equilibria, the results of fluid inclusion studies of the pegmatite and the metamorphic conditions in the area, a constrained P-T path of the magmatic–hydrothermal crystallization of the Jiajika No. 134 pegmatite is proposed. The unusual steeply sloped cooling path of the No. 134 pegmatite could be attributed to the fast pressure drop triggered by the intrusion of a pegmatitic melt along the fractures surrounding the Majingzi granite, which could also be the dominant evolution process for other spodumene pegmatites with similar SQI features in the world. The feature of limited internal geochemical fractionation suggested by mineral-scale geochemical analyses of spodumene and micas, combined with the clear textural zoning of the No. 134 pegmatite, can best be ascribed to the effect of undercooling during pegmatite formation. This effect might be one of the non-negligible rules of pegmatite petrogenesis, and would significantly upgrade the potential of Li mineralization by minimizing diffusional Li transfer to the country rocks.

**Keywords:** spodumene; pegmatite; lithium mineralization; magmatic-hydrothermal evolution; undercooling

## 1. Introduction

The Jiajika rare-metal deposit is the largest lithium deposit in Asia which contains hundreds of pegmatite veins that belong to the LCT family. Most of the world's famous LCT-type rare-metal pegmatites are well zoned both in texture and mineralogy, e.g., Tanco in Canada [1–4] or Koktokay in northwest China [5–7]. However, weakly-zoned rare-metal pegmatites (e.g., Brazil Lake [8], Kings Mountain [9], Leinster pegmatites [10–12]), which are predominantly homogeneous [13] and challenge the general rules of pegmatite petrogenesis [14]. Particularly, confirmed by field-based observations and analysis, the largest lithium pegmatite—No. 134 pegmatite in the Jiajika rare-metal deposit, is characterized by clear zonation in the rock texture and is relatively homogeneous in mineral chemistry among each of the zones. This unique feature is similar to but significantly different from the above types of weakly-zoned pegmatite, and its genesis still remains to be further investigated. Moreover, the magmatic–hydrothermal evolution and the temperature and pressure conditions of the cooling path of the pegmatite, which is closely associated with the Li mineralization of the Jiajika pegmatites, are obscure.

In this study, we carried out detailed petrographic observations and mineral-scale geochemical analyses for each textural zone of the No. 134 pegmatite, and made comparisons among these different zones to investigate the diagenesis and evolution of the pegmatite. This allowed us to decipher the critical controlling factors leading to its unique zonation and super-large-scale Li mineralization, and to obtain a better understanding of the internal evolution of the Jiajika No. 134 rare-metal pegmatite. These new results could help facilitate the exploration of the related granitic pegmatite deposits and optimize prospecting for similar Li ore deposits.

## 2. Geological Background

The Jiajika rare-metal deposit is located in the west of Sichuan Province, situated on the southeastern side of the main body of the Songpan-Ganzê orogenic belt (SGOB) lying on the east margin of the Tibetan Plateau. It is a super-large rare-metal reserve that includes an estimated 1,887,700 t of $Li_2O$ with associated economically variable components of BeO, $Nb_2O_5$, $Ta_2O_5$ and $Cs_2O$ [15,16]. The area of the Jiajika rare-metal deposit covers approximately 62 km$^2$. It is dominated by metasedimentary rocks of the Triassic Xikang Group, which mainly comprise slate, blastopsammite, phyllite, two-mica schists and a range of mica schists with mid- to low-grade metamorphic minerals (e.g., cordierite, tourmaline, andalusite or staurolite). Folds and faults are well developed in this area. The Majingzi two-mica granite pluton, with a total outcropping area of 5.3 km$^2$ in the south-central part of the study area, intruded into the Triassic Xikang Group strata around 223 Ma [16] and caused different degree of thermodynamic contact metamorphism. This led to the formation of a quasi-concentrical progressive metamorphic zonation characterized by diagnostic metamorphic minerals in the strata. The Jiajika pegmatite dikes that intruded into the strata had similar evolution trends to the progressive metamorphic zones in this field, with a zonal distribution in both the horizontal (Figure 1) and vertical directions from the Majingzi granite pluton outwards. Based on their mineral composition, the pegmatite dikes can be divided into five types (Figure 1) with increasing distance from the Majingzi granite pluton, which are: (I) the microcline type, (II) the microcline-albite type, (III) the albite type, (IV) the spodumene type and (V) the lepidolite (or muscovite) type [17].

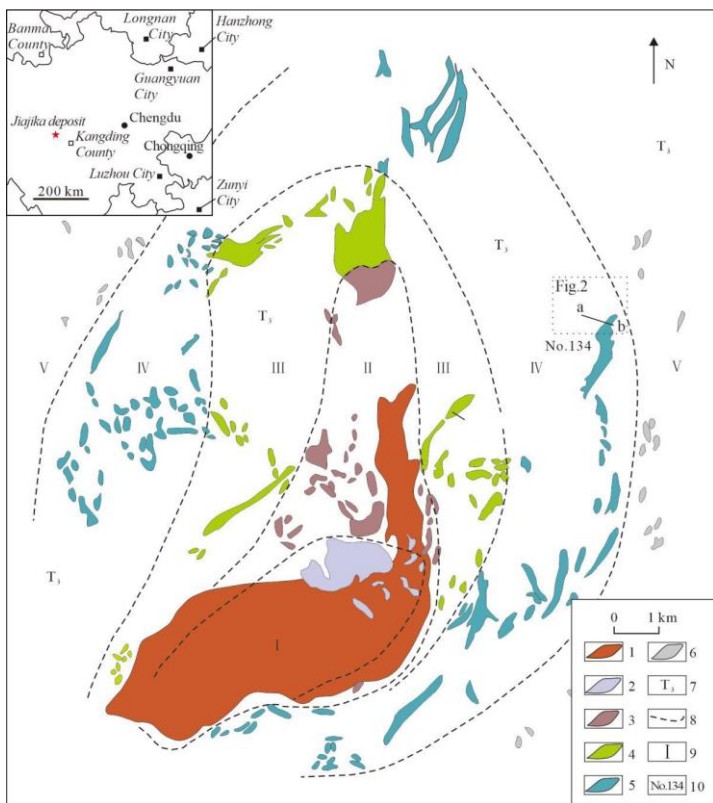

**Figure 1.** Simplified geological map of the Jiajika ore field (modified after Li and Chou [18]). 1—two-mica granite; 2—microcline pegmatite; 3—microcline-albite pegmatite; 4—albite pegmatite; 5—albite-spodumene pegmatite; 6—albite-muscovite(/lepidolite) pegmatite; 7—Triassic system; 8—boundary line of regional zonation of the pegmatites; 9—serial number of regional zonation of the pegmatites; 10—pegmatite dike number. The location of the cross-section of the No. 134 pegmatite dike is shown in Figure 2.

The Jiajika No. 134 pegmatite dike is an albite-spodumene type (Type IV) pegmatite, situated 1.8 km northeast of the Majingzi two-mica granite (Figure 1). According to the muscovite $^{40}$Ar/$^{39}$Ar geochronology of [19], the emplacement of the No. 134 pegmatite occurred at $\pm$ 195.7 Ma. The No. 134 pegmatite dike appears as a lenticular body (Figure 2) striking 24° northeast, and is 20–100 m in thickness and 1055 m in length, with burial depth of 0–200 m [17]. It has a reserve of 512,200 t of $Li_2O$ [19], and, except for several small branches and the pinch-out sides, almost the whole pegmatite body was evaluated as exceeding the economic mineralization value ($Li_2O$ > 1.5%, according to [20]. The No. 134 pegmatite juts out from the ground and forms a protruding hillock, which perfectly meets the requirements for open-pit mining. Although several large-scale rare-metal mineralized pegmatite dikes have been discovered [21] in recent years, the No. 134 pegmatite is still the most economically valuable pegmatite among over a hundred pegmatite dikes in the area and is currently being mined.

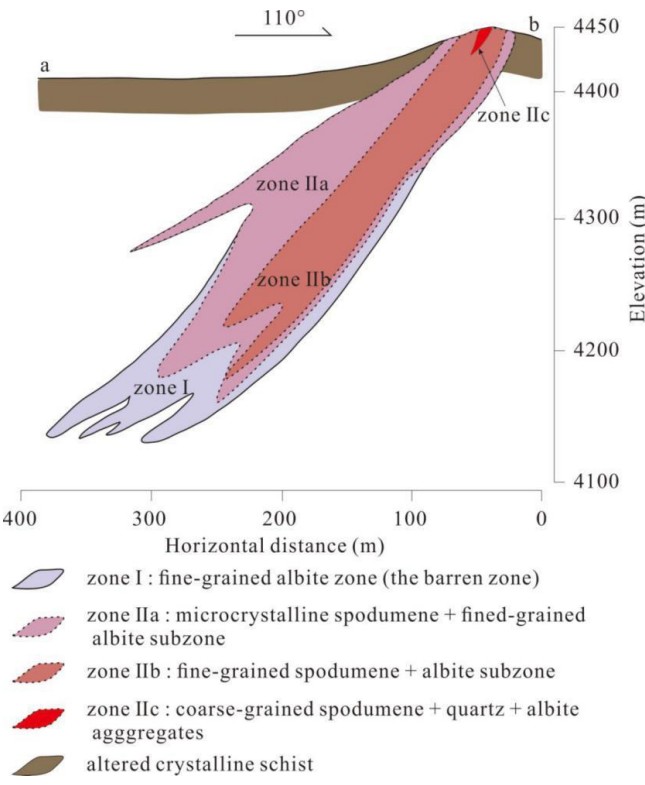

**Figure 2.** Geological section map of the Jiajika No. 134 pegmatite (modified after Liu et al. [22] and Li and Chou [23]).

## 3. Results

### 3.1. Petrographic Characteristics of the No. 134 Pegmatite

The No. 134 pegmatite can be divided into the barren zone (Zone I) and the spodumene zone (Zone II) on the basis of the field observations. The main rock-forming and accessory minerals distributed in each of the textural zones of the Jiajika No. 134 pegmatite are listed in Table 1.

The barren zone constitutes a discontinuous edge of the pegmatite body (Figure 2), which is characterized by large amount of fine-grained albite and the absence of spodumene and, in some cases, the occurrence of beryl.

The spodumene Zone II is the most important textural zone, as it constitutes the bulk of the pegmatite dike, as well as making up the whole orebody of the pegmatite. Based on detailed observations of the exposed section in the quarry, we refined the textural zonations of Zone II into three textural subzones (Zone IIa, Zone IIb and Zone IIc; see below) based on their mineral assemblages and mineral contents. Transitions from one subzone to another are often characterized by diffuse boundaries.

### 3.1.1. Zone I (The Barren Zone: Fine-Grained Quartz–Albite Zone)

Zone I (the barren zone) mainly consists of albite (~50%), anhedral quartz (10–30%), subhedral muscovite (10–20%), beryl (~5%) and K-feldspar (<2%) (Figure 3a). Beryl (<100 μm) is distributed with great variety in its abundance and usually occurs with hydroxylapatite residing within the fractures or coating the crystal rims. Quartz often exhibits undulose extinction and <5 vol% of it appears saccharoidal along the fractures in the rock. K-feldspars are relatively rare and occur as remnant minerals, which can only be observed under SEM (scanning electron microscope) magnification. Accessory minerals include apatite, columbite group minerals (CGM), cassiterite and zircon.

**Table 1.** List of minerals in the Jiajika No. 134 pegmatite and their schematic sequences.

| Lithological Zones | Zone I | Zone IIa | Zone IIb | Zone IIc | Magmatic→Hydrothermal |
|---|---|---|---|---|---|
| K-feldspar | 2 | 30 | 10 | + | |
| albite | 50 | 7–15 | 35 | 20 | |
| quartz | 10–30 | 10–25 | 18 | 25 | |
| muscovite | 10–20 | 5–20 | 10 | 10 | |
| spodumene | | 25 | 20 | 40 | |
| beryl | ++ | + | | | |
| hydroxylherderite | + | | | | |
| CGM | + | + | + | + | |
| lithiophilite | | + | + | | |
| lepidolite | | + | + | + | |
| apatite | ++ | ++ | ++ | + | |
| fairfieldite | | + | + | | |
| rhodochrosite | | + | + | | |
| cookeite | | ++ | ++ | + | |

Notes: Rock-forming minerals were estimated as volume percentages. The elongated quadrilateral represents mineral crystallization.

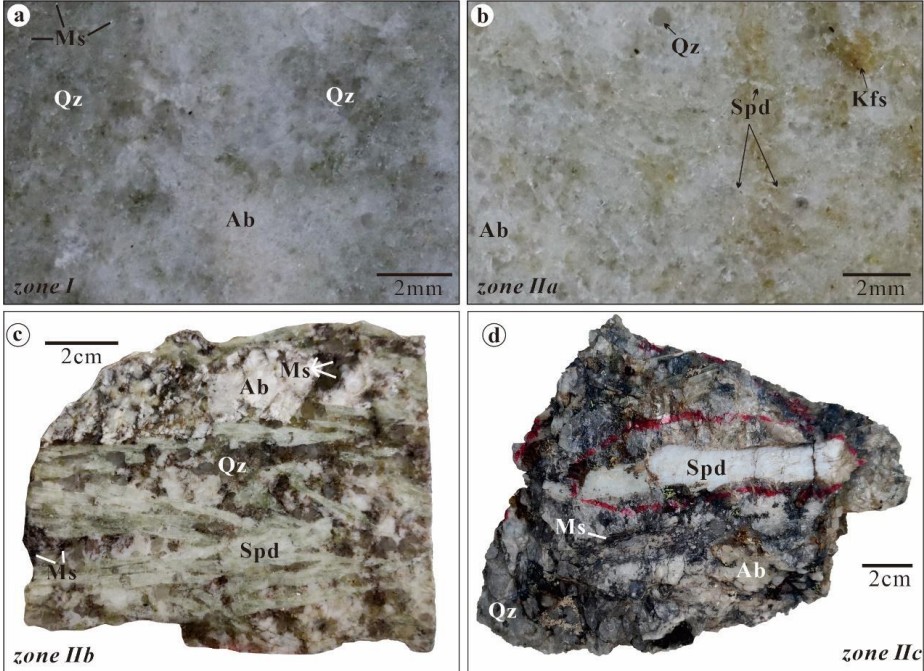

**Figure 3.** (**a**–**d**) Photographs of the different lithologies observed in hand specimens. (**a**) Partial view of a hand specimen from the barren zone (Zone I), showing a very fine-grained structure and containing abundant muscovite, quartz and albite. (**b**) Partial view of a hand specimen of microcrystalline spodumene + fined-grained albite pegmatite (Zone IIa), where spodumene forms short to elongate laths, mostly ranging from 0.1 to 0.2 mm. (**c**) Hand specimen of fine-grained spodumene + albite pegmatite (Zone IIb) with abundant spodumene prisms ranging from 2 to 4 cm. (**d**) Hand specimen of coarse-grained spodumene + quartz + albite aggregates (Zone IIc) containing 10-centimeter-scale spodumene crystals. Abbreviations: Ab: albite, Kfs: K-feldspar, Ms: muscovite, Qz: quartz, Spd: spodumene.

### 3.1.2. Zone IIa: Microcrystalline Spodumene + Fine-Grained Albite Subzone

Zone IIa comprises ~25 vol% subhedral spodumene, 10–25 vol% anhedral quartz, 7–15 vol% albite, 5–20 vol% muscovite and <30 vol% K-feldspar. Microcrystalline spodumene makes short (mostly 0.1–0.2 mm, Figure 3b) to elongated (up to 1 cm, Figure 3b) prisms or laths, often fractured and commonly with zoning patterns (Figure 4b) under SEM. In some cases, symplectic intergrowths of spodumene and albite may occur in the outer ring of a spodumene grain (Figure 4a). Muscovite generally consists of flakes with pronounced cleavage planes, shows chemical heterogeneity and may partly transform into lepidolite when adjacent to spodumene. Unlike in Zone I, beryl is sporadically distributed and shows zoning patterns. Typical accessory minerals are apatite (locally present as fillings in cavities and fractures in rocks and minerals), CGM and cassiterite.

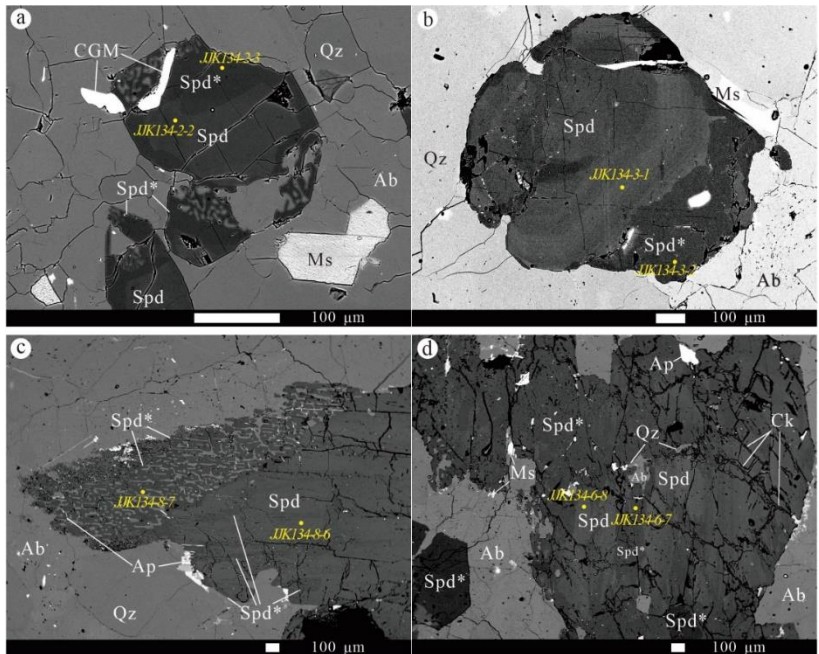

**Figure 4.** (**a–d**) Backscattered electron (BSE) images of lithium minerals of the Jiajika No. 134 pegmatite. (**a**) Microcrystalline spodumene associated with CGMs in Zone IIa, showing symplectic intergrowths with albite at the rim. (**b**) Microcrystalline spodumene in Zone IIa displaying an oscillatory zoned core and a BSE-darker outer ring. (**c**) Fine-grained spodumene in Zone IIb showing symplectic intergrowths with albite at the rim; chemical alteration occurred along its fractures to form a patchy-like zoning. (**d**) Coarse-grained spodumene in Zone IIc, partly altered and replaced. The yellow dot indicates the location of the analysis point, and the corresponding name of the analysis point can be found in Supplementary Table S1. Abbreviations: Ap: apatite; Ab: albite, CGM: columbite group minerals, Ck: cookeite, Ms: muscovite, Qz: quartz, Spd: spodumene, Spd *: secondary spodumene.

### 3.1.3. Zone IIb: Medium- to Fine-Grained Spodumene + Albite Subzone

Zone IIb is dominated by ~20 vol% lath-shaped spodumene (2–4 cm, Figure 3c), ~18 vol% dark gray quartz, ~35 vol% albite, ~10% vol% muscovite and <10% vol% K-feldspar. Spodumene is commonly broken and shows zoning patterns. Cracks and fractures may be filled by cookeite and muscovite, and may contain mineral inclusions such as albite and cassiterite. It also occasionally displays symplectic rims (Figure 4c). Muscovite in the fractures of spodumene may be partly altered to lepidolite along the contact boundary. Accessories include apatite, CGM, cassiterite and xenotime.

### 3.1.4. Zone IIc: Coarse-Grained Spodumene + Quartz + Albite Aggregates

Zone IIc shows coarse pegmatitic mineral assemblages that comprise ~40 vol% centimeter-scale spodumene (6–10 cm, Figure 3d), ~25 vol% dark gray quartz, ~20 vol% albite and ~10 vol% muscovite. Spodumene is often fractured and chemically heterogeneous (Figure 4d); mineral inclusions such as albite, cookeite and muscovite are common. Spodumene crystals may display symplectic spodumene–albite intergrowths at the rim. Muscovite shows very pronounced cleavage planes, and minor flakes of Cs-rich micas may precipitate along those cleavages (e.g., Figure 5c,d). K-feldspars are rarely present. Accessories include apatite, CGM and cassiterite.

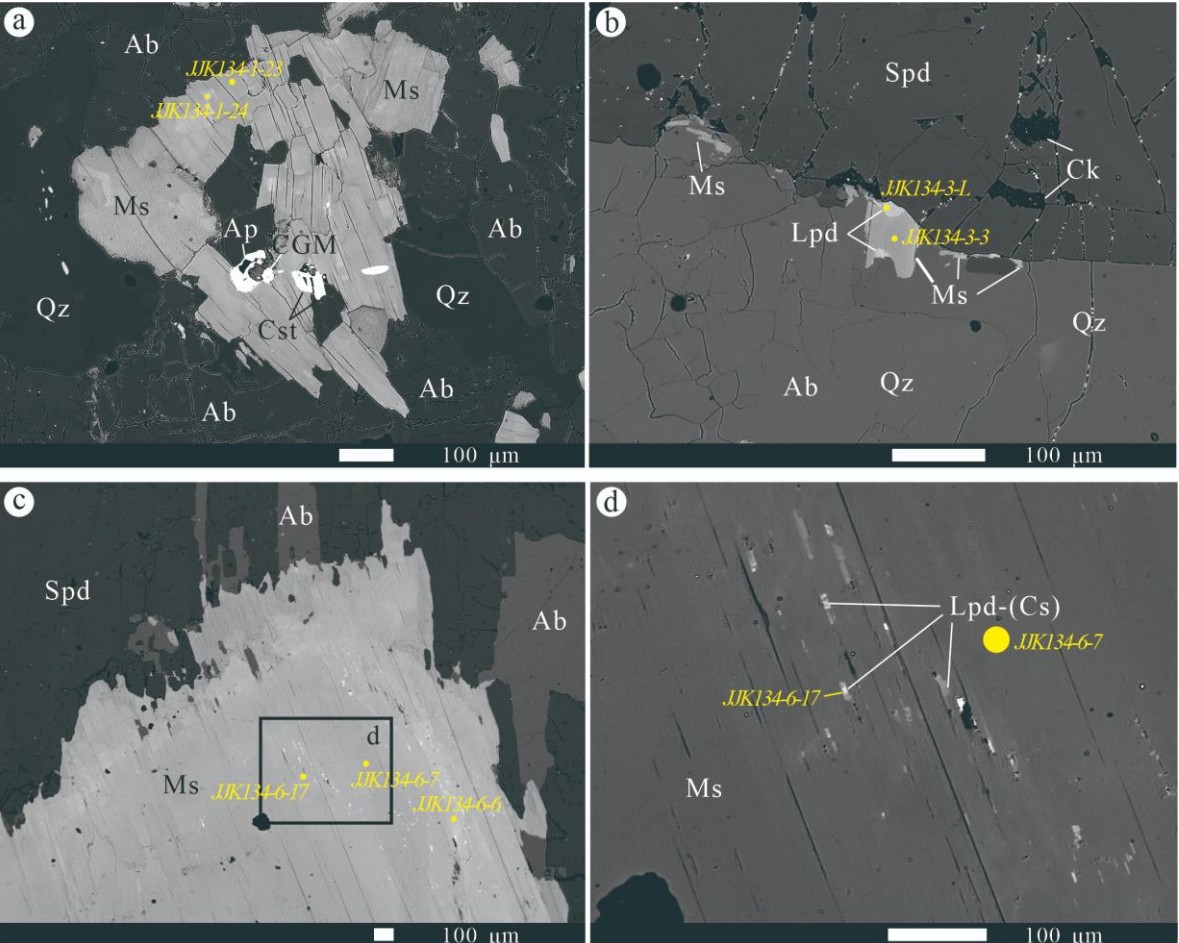

**Figure 5.** (**a**–**d**) Backscattered electron (BSE) images of micas of the Jiajika No.134 pegmatite. (**a**) Muscovite grains in zone I show patchy zoning. (**b**) Muscovite flakes in zone IIa occur along the edge of a spodumene grain, may be partly altered into lepidolite. (**c**,**d**) Minor Cs-rich mica flakes precipitated along the cleavage planes of one giant patchy zoning muscovite in zone IIc. The yellow dot indicates the location of the analysis point, and the corresponding name of the analysis point can be found in Supplementary Table S2. Abbreviation: Ap: apatite; Ab: albite, CGM: columbite-group minerals, Ck: cookeite, Cst: cassiterite, Lep: lepidolite, Ms: muscovite, Qz: quartz, Spd: spodumene.

The mineral chemistry of silicates was determined under the operating conditions of an accelerating voltage of 15 kV, a beam current of 20 nA and a beam diameter of 10 μm for micas but 5 μm for all other minerals. The determination time of the characteristic peaks of Cs, Sr, Nb and Ta was set as 20 s; that of the oxides of other elements was set as 10 s and that of all oxide backgrounds was set as 5 s. Quantitative analyses of the minerals were calibrated using the following natural and synthetic standards: quartz (Si-*Kα*), jadeite (Al-*Kα* and Na-*Kα*), wollastonite (Ca-*Kα*), apatite (P-*Kα*), phlogopite (K-*Kα*, F-*Kα*), forsterite

(Mg-*Kα*), TiO$_2$ (Ti-*Kα*), MnTiO$_3$ (Mn-*Kα*), Fe$_2$O$_3$ (Fe-*Kα*), spinel (Cr-*Kα*), cassiterite (Sn-*Lα*), lithium niobate (Nb-*Lα*), lithium tantalate (Ta-*Mα*), pollucite (Cs-*Kα*), NaCl (Cl-*Lα*), coffinite (U-*Mα*, Th-*Mα*) and PbCr$_2$O$_5$(Pb-*Mα*). The detection limit for most elements during this EPMA analysis was $40 \times 10^{-6}$–$200 \times 10^{-6}$. ZAF correction was applied.

### 3.2. Mineralogical Characteristics of the No. 134 Pegmatite

Optical microscopy and EPMA analyses revealed a variety of rare-metal carrier minerals in the No. 134 pegmatite. In this study, we focused on spodumene, mica and beryl.

### 3.2.1. Spodumene

Spodumene occurs as one of the main rock-forming minerals in the Jiajika No. 134 pegmatite and could be easily recognized in the pegmatite outcrops through field observation. It is absent in Zone I but distributed with variable abundance and in different occurrences in Zone II. Petalite relicts are rarely observed in the symplectic rim of spodumene, and cookeite commonly occurs as an alteration product of primary spodumene. A clear observation of these minerals could be obtained under the microscope.

For each subzone of Zone II, spodumene crystals show these similar characteristics: they are commonly broken, with hydroxylapatite intersecting the crystals along their fractures in some cases; they may display symplectic spodumene–albite intergrowths at the rim; and they may show zoning patterns resulting from slight changes in elemental contents under a SEM (scanning electron microscope).

In Zone IIa, microcrystalline spodumene usually shows simple radial zoning, consisting of a BSE-bright inner core and a BSE-dark outer ring separated by a sharp interface (Figure 4a), which could be best ascribed to the replacement process along the interface through the almost coeval occurrence of dissolution and reprecipitation [24,25]. In rare occurrences, the core of the radial zoned crystals displays oscillatory zoning, characterized by very thin individual zones parallel to the crystal edges, suggesting a magmatic origin (Figure 4b). Symplectic intergrowth of spodumene and quartz may discontinuously occur along the edge of the spodumene crystals in all the subzones of textural Zone II, and becomes a major textural feature of the spodumene in Zones IIb and IIc, where coarser spodumene crystals dominate. Additionally, the zoning patterns of spodumene crystals in Zone IIb also include simple radial zoning, which is similar to that of Zone IIa.

The chemical composition of spodumene varies but is generally close to the ideal formula, LiAlSi$_2$O$_6$. The calculations yielded high molecular totals (average values of up to 103.0%; see Supplementary Table S1) but were still within acceptable error limits, considering the uncertainties in the determination of Li. The results show that the chemical composition of spodumene in each subzone of Zone II varies irregularly; however, within individual zoned spodumene crystals, an obvious pattern of the chemical changes can be concluded as follows: from the crystal core to the outer ring and/or the rim in the zoned crystals, femic components (FeO + MnO) decrease and Li$_2$O contents increase. Generally, the altered rim of a spodumene crystal contains about 0.1 wt.% more Li$_2$O than its core. However, this law is not applicable to a comparison between different crystals; there is no chemical justification for distinguishing rim versus core chemistry in spodumene. Other components including Fe and, to a lesser extent, Sr, Na and Mn, occur as small amounts of impurities in the chemical composition of spodumene (Supplementary Table S1).

### 3.2.2. Other Lithium Minerals

Cookeite usually crosscuts spodumene along its {110} cleavages without affecting its chemical zoning (Figure 4d), resulting in enlarged cleavages. This observation reflects the scale of local micro-environmental control of spodumene alterations [26]. The transition from spodumene to cookeite took place in the late stage of the evolution of aluminosilicate minerals and the pegmatite, where an acid fluid probably emerged in response to the melt–fluid interaction (5 Spd + 8 H$^+$ → Ck + 7 Qtz + 4Li$^+$; [12,27]). The cookeite is moderately

enriched in Si (34.63–39.07 wt.%) and poor in Al (44.05–46.70 wt.%) relative to the ideal formula of $LiAl_4(Si_3Al)O_{10}(OH)_8$.

Lithiophilite is a Mn-rich member of the triphylite–lithiophilite series and usually crystallizes at a late stage of crystallization of primary lithium minerals in pegmatites [28–31]. Here in the Jiajika No. 134 pegmatite, lithiophilite is the only primary lithium phosphate mineral present. They are rarely preserved, usually appearing as tiny relicts (<5 μm) included in apatite (and in fairfieldite, in a few cases), suggesting that the primary lithiophilite had undergone a thorough alteration processes. Lithiophilites in different textural zones do not show significant variations and they have an average Fe/(Fe + Mn) ratio of 0.39, corresponding to an Mn-rich member of the triphylite–lithiophilite series, and low Mg contents of <0.002 Mg apfu. The incipient crystallization of lithiophilite rather than triphylite probably indicates that the pegmatite was highly differentiated with respect to Mn vs. Fe when it was emplaced, possibly originating from a highly fractionated granitic source [28].

### 3.2.3. Micas

Mica minerals are widespread in all textural zones of the Jiajika No. 134 pegmatite. They are distributed in each zone in a similar range (10–17%). From Zone I inwards, the mica flakes and laths vary from very fine (<10 μm) to coarse (>5 cm) as a result of the different formation conditions.

In Zone I, muscovite is either homogeneous or shows irregular patchy zoning (Figure 5a); such features are common in other textural zones as well (e.g., Figure 5c). For the muscovite crystals with patchy zoning, the BSE-bright and BSE-dark parts generally reflect differences in the major element contents (Figure 5a,c); for instance, patchy-zoned muscovite from Zone IIc shows high femic contents (3.04–3.10 wt.%) and lower $Al_2O_3$ contents (33.97–36.28 wt.%) for the BSE-bright parts, and lower femic contents (2.14–2.2 wt.%) and higher $Al_2O_3$ contents (35.24–35.38 wt.%) for the BSE-dark parts.

For each subzone of Zone II, small flakes of mica may occur discontinuously along the edge of spodumene grains (Figure 5b), indicating a secondary origin. Li-mica is rare but is present in every textural zone of Zone II, though it only occurs as alteration product of primary muscovite. In Zone IIa, Li-mica may occur as discontinuous rims of muscovite which contacts spodumene (Figure 5b); in Zone IIb, it occurs as rims of muscovite set in fractures of spodumene (e.g., Zone IIb). Notably, in Zone IIc, groups of micrometer-scale flakes of Cs-rich mica may appear along the cleavage planes of the primary muscovite grains (Figure 5c,d), the highest $Cs_2O$ content of which can reach 16.8 wt.% (Supplementary Table S2).

### 3.2.4. Beryl

Beryl is the only primary Be-bearing mineral in the No. 134 pegmatite. Beryl crystals are ubiquitous in Zone I but rarely present in other zones; a few were observed in samples from Zone IIa and Zone IIb. In Zone I, beryl grains are usually discretely distributed in muscovite and albite (Figure 6a,b). Hydroxylapatite often occurs as a secondary phosphate residing within the fractures or coating the rim of beryl crystals (Figure 6a). Hydroxylherderite is immersed within these crosscutting veinlets (Figure 6b). In Zones IIa and IIb, beryl grains are often associated with spodumene and muscovite (e.g., Figure 6c,d). Beryl in Zone II commonly shows compositional heterogeneity, as it may display alterations on the rim or form patchy zonation (Figure 6c,d), while beryl in Zone I is mostly homogeneous and, in a few cases, very weakly zoned.

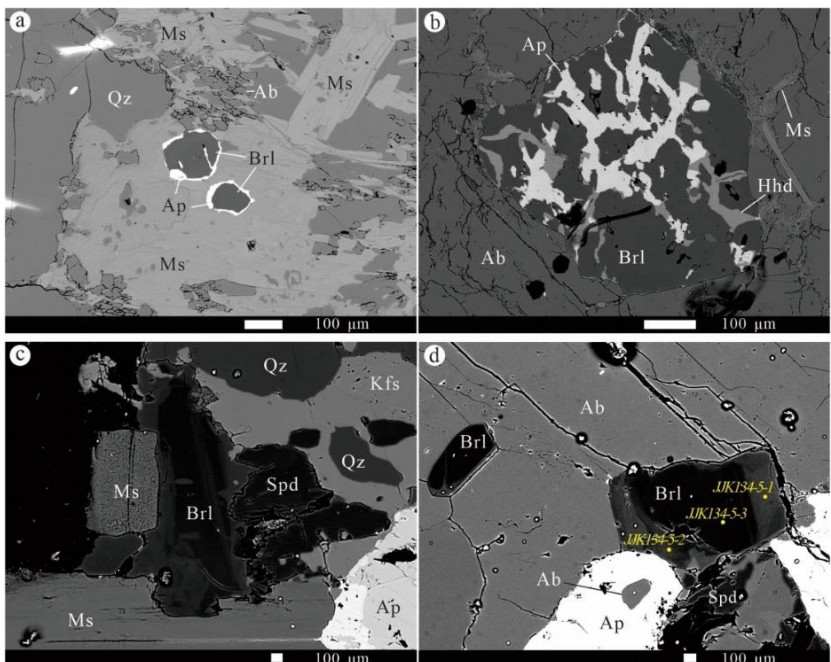

**Figure 6.** (**a**–**d**) Backscattered electron (BSE) images of beryls from the Jiajika No. 134 pegmatite. (**a**) Beryl grains included in muscovite in Zone I, with hydroxylapatites rimming the edge. (**b**) Beryl grains included in albite in Zone I, with hydroxylherderite and hydroxylapatite intergrowths. (**c**) Compositionally heterogeneous beryl grains associated with spodumene and muscovite in Zone IIa, with elevated $Cs_2O$ and $Li_2O$ content on the rim. (**d**) Beryls in Zone IIb display patchy zoning on the rim. The yellow dot indicates the location of the analysis point, and the corresponding name of the analysis point can be found in Supplementary Table S3. Abbreviations: Ap: apatite; Ab: albite, Brl: beryl, CGM: columbite group minerals, Hhr: hydroxylherderite, Kfs: K-feldspar, Ms: muscovite, Qz: quartz, Spd: spodumene.

Electron microprobe analyses of beryl grains of the No.134 pegmatite indicate that except for the outer ring of these patchy-zoned beryl crystals, which may represent alteration by an exsolved fluid, the primary beryls in Zone I and Zone II are chemically similar. No obvious disparity in chemistry was observed between these different zones (Supplementary Table S3). However, the result of the chemical composition of individual beryl grains reveals the trend of an increase in alkali components and a decrease in femic components from the primary to secondary phases. The chemical composition is similar to that of primary magmatic beryl crystals from the Nanping No. 31 pegmatite at Fujian [32] and is comparable with the primary and secondary beryl grains from the Koktokay No. 3 pegmatite at Xinjiang [7].

## 4. Discussion

### 4.1. Li-Saturated Pegmatite-Forming Melt

The typical graphic pegmatite zone is absent in the Jiajika No. 134 pegmatite; the very thin and discontinuous border zone contains almost no lithium minerals, and spodumene crystallizes throughout nearly the whole pegmatite body. The rock-forming minerals are invariable in species (K-feldspar, albite, quartz, muscovite and spodumene) but moderately varied in contents (Table 1) and are similar in mineral chemistry while distinctly variable in grain size. These features suggest a well-zoned but relatively chemically homogeneous pegmatite.

The No. 134 pegmatite shows extreme Li enrichment coupled with a relatively simple mineralogy, which may not be consistent with the equilibrium crystallization of a granitic melt [33]. Experimental studies showed that rather than occurring at once when the saturation point has been reached, the crystallization of minerals from the melt in

pegmatites always commences after a necessary lag time for crystal incubation [34]. In Li-bearing pegmatite melts, large amounts of Li competing with Na and K for appropriate polymers in mineral formation would consequently make any of the competing minerals incapable of nucleating [35,36]. Even more specifically, the results of the experimental study of Maneta et al. [37] showed that the nucleation of Li-aluminosilicates is significantly delayed, and that Li-supersaturated pegmatite-forming melts might have migrated for several kilometers [38] before crystallizing any lithium minerals and would precipitate a considerable abundance of Li-aluminosilicate in proximity to the surrounding wall rocks after rapid cooling of the melt [37,39,40]. Moreover, Li saturation before crystallization would probably lead to simultaneous nucleation of spodumene throughout the whole pegmatite magma body [12]. The No. 134 pegmatite shows the characteristics of a very thin and discontinuous Li-free margin and randomly oriented spodumene crystals distributed throughout almost the whole pegmatite body, which is consistent with the results of these aforementioned studies [12,37,40]. This evidence strongly suggests Li saturation prior to crystallization for the system.

### *4.2. Evolution of the Jiajika No. 134 Pegmatite*

#### 4.2.1. Magmatic–Hydrothermal Evolution

Variations in the mica species and specific element content and/or ratios of mica and beryl may indicate the crystallization history of minerals, which could provide constraints on the nature and the processes of petrogenesis and mineralization for the system, considering that such minerals are regarded as key indicators for understanding the evolution of granitic systems [11,41–45].

The beryllium speciation in a granitic melt is mainly controlled by the competition between Si and P as network-forming cations in the melt [46]; thus, the existence of beryl, combined with the presence of primary spodumene and quartz, indicates a Si-saturated crystallizing environment. Previous studies concluded that the average content of alkali and femic metals (such as Cs, Li and Fe) and certain element ratios (such as Na/Cs) in beryl may reflect the genetic and geochemical features of the crystallization media [7,47,48]. From Zone I to Zone IIa and to Zone IIb in the Jiajika No. 134 pegmatite, the contents of the femic components in beryl gradually decrease as the contents of the alkaline components increase (Figure 7a), and the $Cs_2O$ content increases with the increase in $Na_2O$ content from the border zone inwards (Figure 7b). Such variation was also observed in individual crystals with zoning patterns in Zones IIa and IIb, in which the variation in $Cs_2O$ content is the most significant (up to two orders of magnitude). The $Cs_2O$ content of primary beryl in the No. 134 pegmatite varies from 0–3.57%, which is comparable with that of the primary beryl in Koktokay (0–3.79% from the border zone to textural Zone VII, [7]), the primary beryl in Nanping No. 31 pegmatite (0–3.52% from Zone I to Zone IV, [32]), the primary beryl in Tanco pegmatite (maximum $Cs_2O$ 3.79%, [1]) and the beryl in Bikita pegmatite (maximum $Cs_2O$ 3.79%, [47]. As the incompatible Cs always accumulates during magma evolution, such comparability in $Cs_2O$ content indicates the similarity of Cs enrichment in the initial melt and during the magmatic evolution of the Jiajika No. 134 pegmatite to other important rare-metal pegmatites in the world, which indicates the high degree of fractionation of the Jiajika No. 134 pegmatite.

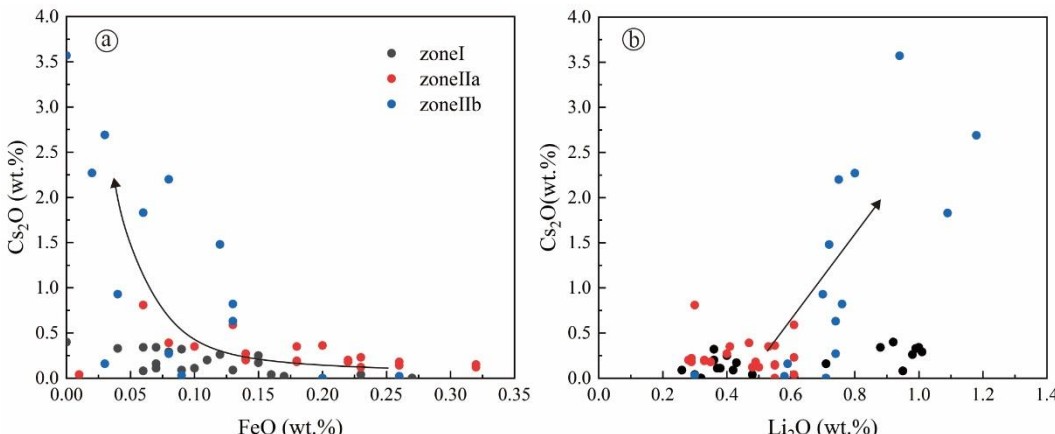

**Figure 7.** Chemical variation of (**a**) $Cs_2O$ vs. FeO and (**b**) $Cs_2O$ vs. $Li_2O$ in beryl from each zone of the Jiajika No. 134 pegmatite.

The evolution of mica species from muscovite to Li-mica from Zone I to Zone II also supports the identifiable magmatic evolution of the No. 134 pegmatite from the outer zone inwards. The $Li_3Al_{-1}\square_{-1}$ and $Al_2\square(R^{2+})_{-3}(R^{2+} = (Fe + Mg + Mn)$ mechanism operates in the transition from dioctahedral to trioctahedral micas, which is common in the individual mica crystals of all the subzones of textural Zone II (Figure 8). Such chemical changes document the transition from the magmatic stage to a post-magmatic stage, and can be directly reflected by the heterogeneity of the gray values of mica minerals under SEM. This transition indicates that a change in the crystallization environment towards one with favorable conditions for precipitating secondary Li-mica took place in the formation process of Zone II, and the change in the crystallization environment was probably generated in the presence of an exsolution fluid with high $H^-$ and $F^-$ activity [27].

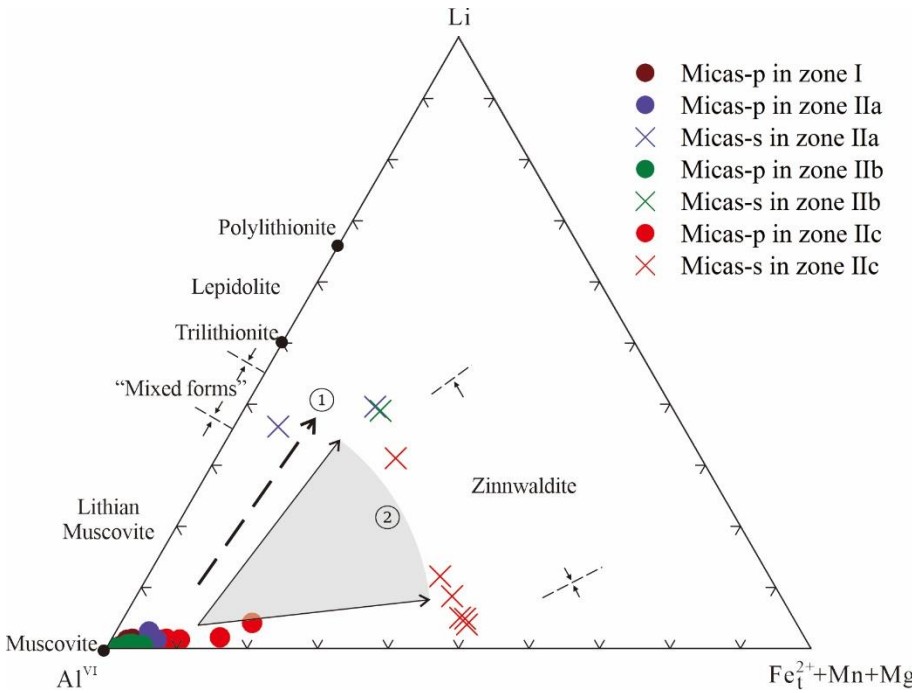

**Figure 8.** Ternary diagram of mica mineral types in different structural zones of the No. 134 pegmatite (base map from Jolliff et al. [41]; Neiva [49]). ① The composition of muscovite→lepidolite is mainly replaced by $Li_3Al_{-1}\square_{-1}$; ② Muscovite→Cs-rich mica with $Al_2\square(R^{2+})_{-3}$ and $Li_3Al_{-1}\square_{-1}$ substitutions ($R^{2+} = (Fe + Mg + Mn)$). Micas-p, primary micas; Micas-s, secondary micas.

However, the contents of $Li_2O$, F and $Rb_2O$ in primary muscovite in each subzone of textural Zone II vary over similar ranges (Supplementary Table S2), and the trends of Li, F, Rb and Cs versus the K/Rb ratio show semblable patterns (Figure 9c–f), illustrating a similar crystallization environment for primary mica in all subzones of textural Zone II. Except for the Cs-rich mica with extremely high $Cs_2O$ contents in Zone IIc (the highest $Cs_2O$ content could reach 16.8 wt.%), the secondary Li-mica in different textural zones shows similar variations in $Cs_2O$, $Li_2O$, F and $Rb_2O$ contents (Figure 9a,b,e,f). Both from crystal to crystal within Zone I and Zone II and inside individual crystals in all subzones of textural Zone II, the contents of Li, F and Cs in mica minerals tend to increase as the K/Rb ratios drop (Figure 9), which follows the normal trend of variation in mica chemistry as pegmatite evolves [43,50–54]. These results all suggest that the system evolved from the magmatic stage (Zone I) to the magmatic–hydrothermal stage (Zone II), and the system had gone through fluid exsolution (fluid with high F, Li and Cs) during the formation of textural Zone II. Moreover, the lack of individual lepidolite crystals, combined with the relatively limited scale of alteration in spodumene, indicates that the activity of the hydrothermal fluids in the system is limited. As hydrothermal alteration of Li minerals always causes Li leaching, we concluded that the limited activity of the hydrothermal fluids would possibly have upgraded the ore-forming potential of the No. 134 pegmatite.

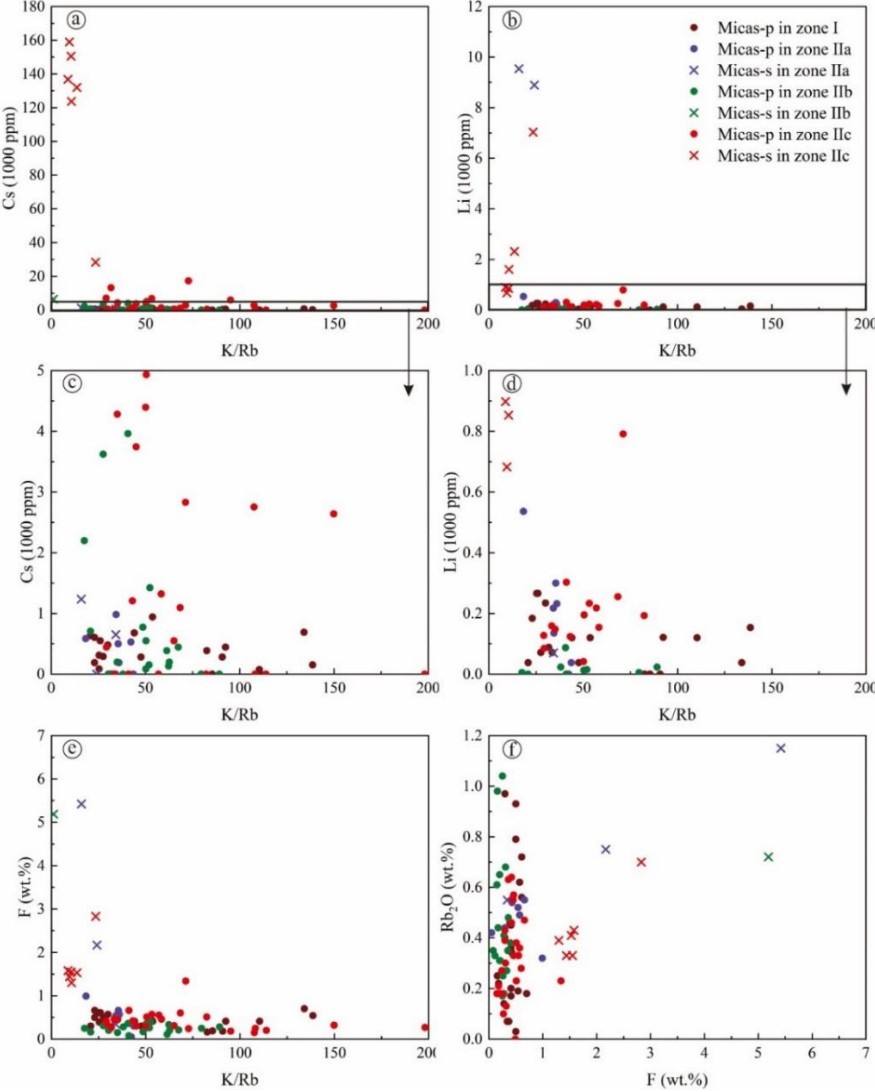

**Figure 9.** Plots of (**a**,**c**) Cs vs. K/Rb, (**b**,**d**) Li vs. K/Rb, (**e**) F vs. K/Rb and (**f**) $Rb_2O$ vs. F in micas from each zone of the Jiajika No. 134 pegmatite. Micas-p, primary micas; Micas-s, secondary micas.

### 4.2.2. P-T Conditions

The stability relationships of the lithium aluminosilicate minerals eucryptite, spodumene and petalite under subsolidus conditions in a $LiAlSiO_4$-$SiO_2$-$H_2O$ system can be shown in the Li-Al silicate grid proposed by London [55]. The moderate- to low-T proportion of this grid is applicable to the petrogenesis of Li-rich pegmatites. Whether the Li-rich pegmatites first crystallized petalite and then spodumene (e.g., Harding pegmatite [56]. Tanco pegmatite [57], Highbury pegmatite [58]) or directly crystallized spodumene (e.g., the aplite–pegmatite dykes of the Covas de Barroso district in northern Portugal [59] and Leinster pegmatites [12]) as the primary lithium aluminosilicate depends on the external P-T conditions. In the latter case, the petalite is commonly isochemically replaced by spodumene + 2 quartz (spodumene–quartz intergrowths, commonly referred to as "SQI") [57,58] or remains metastable in very rare cases [59] as the P-T conditions change.

During the magmatic stage of evolution in the Jiajika No. 134 pegmatite, spodumene was the first Li-bearing aluminosilicate to crystallize from the melt (500–700 °C) [60], which marked the onset of Li mineralization. Spodumene–quartz intergrowths usually occur partly along the rims of the spodumene grains and do not penetrate or are not penetrated by any primary spodumene grains in the Jiajika No. 134 pegmatite. This texture indicates that the spodumene–quartz intergrowths formed after the crystallization of primary spodumene. Similar observations have been reported around the world: in the Vredefort spodumene pegmatite in South Africa, petalite crystallized after spodumene under an elevated temperature provided by post-impact heating [61], and petalite precipitated directly from a late orthomagmatic fluid where spodumene was an early magmatic phase in some aplite–pegmatite dykes of the Covas de Barroso district, northern Portugal [59]. In contrast to these cases, the petalite in the Jiajika No. 134 pegmatite has almost entirely been replaced by spodumene–quartz intergrowths, and the occurrence of SQI is quite limited (only as discontinuous rims). These features suggest that in the evolutionary history of the aluminosilicate minerals in the Jiajika No. 134 pegmatite, (1) spodumene crystallized as an early aluminosilicate, (2) petalite precipitated during the change in P-T conditions during late magmatic stage of evolution, and (3) the P-T conditions then returned to the spodumene stability field, and petalite decomposed into SQI. In fact, in other lithium pegmatites, such as the Koktokay No. 3 pegmatite [54] and the Bailongshan No. 2 pegmatite [27] in Xinjiang, China, and the Leinster spodumene pegmatite in southeast Ireland [12], the texture of spodumene–quartz intergrowths formed by the decomposition of petalite occurs locally along the edge of primary spodumene, as occurs in Jiajika. This occurrence shows that petalite precipitation after spodumene is not a rare phenomenon, and there should be a common process controlling its formation in Li-rich pegmatites.

Therefore, we proposed an inferred P-T path for the magmatic–hydrothermal crystallization of the Jiajika No. 134 pegmatite constrained by (1) pegmatite mineral equilibria; (2) the P-T conditions in spodumene crystals of the Jiajika pegmatites measured by fluid-inclusion studies [18,62] and (3) the conditions of metamorphism of the surrounding host-rocks [17]. An experimental study by London [63] showed that pegmatite in which spodumene is the primary Li-aluminosilicate is constrained to crystallization pressures in the range of 300 to 400 MPa at magmatic temperatures of 500 to 700 °C (Figure 10, Condition 1), and the stability of spodumene + quartz (without petalite) represents a minimum pressure of 250 to more than 300 MPa at 450 to 550 °C (Figure 10, Condition 2). Eucryptite has not yet been found in the Jiajika No. 134 pegmatite, and thus 170 MPa could be used as the minimum pressure limit for the P-T conditions. Li and Chou (2016) proposed that the stability pressure of spodumene + quartz is higher than ~440 MPa at 500 to 700 °C (Figure 10, Condition 3). An experimental study by Xiong [62] showed that the initial melting temperature of fluid inclusions with daughter minerals hosted in spodumene ranges from 500 to 580 °C and that the calculated lower limit of the metallogenic pressure is 310 to 480 MPa (Figure 10, condition 4) for the Jiajika pegmatites. Spatio-temporal relationships [64] indicated that the No. 134 pegmatite is younger than the nearby granites, and the metamorphic and deformed surrounding host rock; therefore, the peak temperature and

pressure conditions experienced during host rock metamorphism [16,17,19] may constrain the upper temperature and pressure limits of pegmatite's emplacement and crystallization.

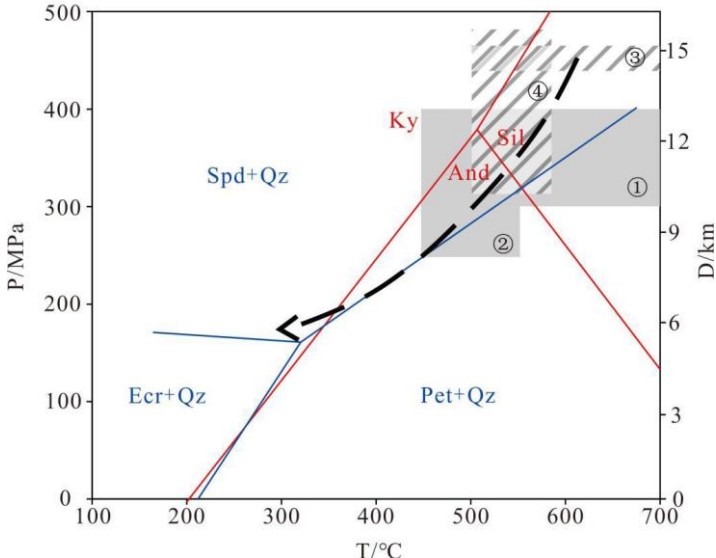

**Figure 10.** Inferred P-T path (arrow with the dashed line) of the magmatic–hydrothermal crystallization of the Jiajika No. 134 pegmatite. The blue line represents the lithium aluminosilicate phase transition line [65]; the red line represents the three-phase point of an aluminosilicate system [23]; the gray area is the limiting Conditions 1 and 2, and the diagonally striped area is the limiting Conditions 3 and 4. See the text of this section for the meanings.

On the basis of this evidence, we constrained a cooling path characterized by a relatively fast drop in pressure with respect to temperature at >~400 °C, whereas the temperature drops faster than the pressure at lower temperatures, as shown in Figure 10. Compared with the common shallow-sloped cooling paths of other lithium pegmatites [55,58], the steeply sloped cooling path of the No. 134 pegmatite is distinctive, which could be attributed to the fast pressure drop that resulted from the intrusion of the pegmatitic melt along the fracture surrounding the granite, driven by multiple magmatic events in the region [17]. This cooling path, which represents the crystallization of the Jiajika No. 134 pegmatite, could also indicate the process of evolution of other spodumene pegmatites with similar SQI features around the world.

### 4.3. Effect of Undercooling on Rock Texture

The internal zonation of pegmatites is manifested both in mineralogy and in rock texture, and has been a focal point in understanding the petrogenesis of pegmatites [66,67].

From the outer to the inner zones of the No. 134 pegmatite, the spodumene crystals vary from microcrystalline (~100 μm) to coarse (>10 cm). Since spodumene was probably formed by simultaneous crystallization, and the pegmatite shows limited internal fractional crystallization, considering the large volume of all the pegmatite, the large size difference of the minerals can best be ascribed to the different magnitude of liquidus undercooling [68–73]. The term undercooling (ΔT) refers to the difference between the liquidus temperature of a mineral and the actual temperature of magma. In undercooled systems, the Gibbs free energy produced by the difference between the growing minerals and the melt is large and negative, providing a significant driving force for crystal growth [36]. Undercooling could exceed the equilibrium crystallization conditions and become the main influencing condition for mineral crystallization and exsolution of a volatile phase in highly undercooled pegmatite liquids [36]. Retarded crystal nucleation and kinetic effects in the melt, including nucleation density, the nucleation rate, the crystal growth rate and the diffusion of mineral constituents, were significantly constrained by the effects of

undercooling on the crystallization response of the pegmatite liquid [36,67]. In general, the nucleation and growth rates of minerals both increase as the ΔT increases and start to drop after reaching their maximum at a certain ΔT (Figure 11; [74,75]). In pegmatites, the fine-grained outer zones represent the initial undercooled crystallization of relatively highly undercooled and dry systems that formed as the consequences of high nucleation density and rapid growth [72], and the heat of crystallization would subsequently moderate the dramatic temperature decrease, such that the residual liquid would probably crystallize at a higher temperature than the liquid at the margins. Evidence that could support this assumption has been reported in the literature; for example, primary inclusions in the wall zone of Animikie Red Ace pegmatite suggest fluid-trapping temperatures at ~480 °C, which is at ~240 °C of undercooling, colder than that in the postdated secondary inclusions, which lie between 580–720 °C, representing fluid exsolution from the inner zones [76]. Likewise, primary inclusions in the core zones of the San Diego dikes suggest colder fluid-trapping temperatures in the wall zones than in the intermediate zones, and that temperatures in the wall zones and the core and pocket zones are similar [36]. These results delineated a typical trend of the undercooling of pegmatites, i.e., beginning at a very large ΔT as a result of emplacement into colder wall rocks, which resulted in high nucleation density and a relatively high rate of crystal growth; the ΔT then rapidly dropped to its minimum owing to the heat produced by crystallization and the large volume of the residual melt, and continued to increase as the melt crystallized towards the pegmatite core, which led to an even more rapid crystal growth rate and a lower nucleation density (as explained below). This trend of undercooling is consistent with the crystallization of a conductively cooled pegmatite body.

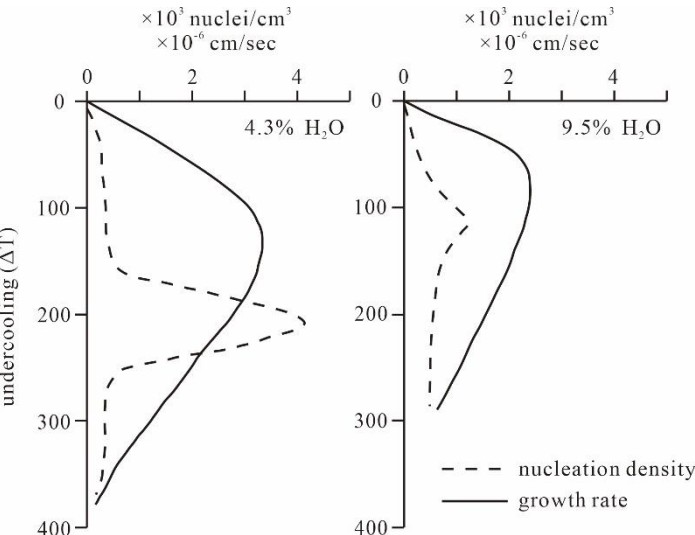

**Figure 11.** Nucleation densities and growth rates as a function of undercooling (ΔT) in an alkali feldspar melt, according to Fenn [74] and Nabelek et al. [36].

Although the effect of $H_2O$ in reducing nucleation density [36,72,77] and enhancing crystal growth [69,71–73,78] has been proved by numerous studies, the crystallization that starts from small ΔT values could, however, result in very limited mineral nucleation (i.e., at a very low nucleation rate) regardless of the $H_2O$ content. In this case, the crystal growth rates would rapidly increase (Figure 11) as the pegmatite gradually cools (ΔT becomes larger), while the nucleation rates keep being decreased by rival cations during crystallization [79]. Nevertheless, the role of $H_2O$ and fluxes such as F, P and Li, which accumulate during crystallization, are not negligible in promoting the morphologic features of pegmatites, as they could delay mineral nucleation to large ΔT values, lower the silicate melt viscosity and enhance the diffusion of the mineral constituents [80–84]. Especially, in comparison with the Li-free system, the presence of Li in the hydrous granitic system

significantly lowers the degree of undercooling (e.g., by approximately 50–70 and 200 °C with the addition of ~3700 ppm Li) needed for crystallization and melting temperatures (Meneta and Baker, 2014), which provides an indication of the crystallization of the Li-rich pegmatites at a relatively low ΔT. Therefore, except that the thin and discontinuous border zone (Zone I) formed under highly undercooled conditions, the crystallization of the main body of the No. 134 pegmatite probably started from a small ΔT and thus led to the inward coarsening of the crystals. Hence, it can be concluded that the features of the relatively compositionally homogeneous but clearly texturally distinguishable zonations of the No. 134 pegmatite can be best explained by rapid crystallization starting from a low degree of undercooling of a Li-supersaturated melt.

Moreover, rapid cooling would significantly contribute to the preservation of Li in the pegmatite during its emplacement [85], since Li tends to transfer from pegmatite to country rocks by diffusion [86,87]. Thus, we concluded that rapid cooling of the pegmatite, despite the other factors such as the degree of crystal fractionation [88] and the activity of hydrothermal fluids, was also an important controlling factor of the ore-forming process of the pegmatite.

## 5. Conclusions

The unique textural zonation of the No. 134 pegmatite was probably generated by the effects of undercooling, which led to the seemingly contradictory attributes of clear textural zonation and limited geochemical fractionation between the textural zones of the pegmatite.

The characteristic Li-enrichment of the pegmatite-forming melt, the rapid cooling rate of crystallization after pegmatite emplacement and the limited-scale hydrothermal activity during the process of the magmatic to the hydrothermal stage of pegmatite evolution are three important controlling factors in the ore-forming potential of the Jiajika No. 134 pegmatite.

A well-constrained P-T path for the magmatic–hydrothermal crystallization of the Li-aluminosilicates in Jiajika No. 134 pegmatite is proposed, suggesting that the intrusion of the pegmatitic melt along the fracture resulted in a fast pressure drop in the region during pegmatite crystallization, and could also provide information on the process of evolution for other spodumene pegmatites with similar SQI features around the world.

**Supplementary Materials:** The following are available online at https://www.mdpi.com/article/10.3390/min12010045/s1, Table S1. EPMA compositions of spodumene in different texture zones in the Jiajika No. 134 pegmatite, Table S2. EPMA compositions of mica in different texture zones in the Jiajika No. 134 pegmatite, Table S3. EPMA compositions of beryl in different texture zones in the Jiajika No. 134 pegmatite.

**Author Contributions:** Conceptualization, Z.W.; Data curation, Z.W. and Z.C.; Formal analysis, Z.W.; Funding acquisition, J.L. and Z.C.; Investigation, Z.W., J.L., Z.C., Q.Y., X.X., P.L. and J.D.; Methodology, Z.W., J.L., Z.C., Q.Y., X.X. and P.L.; Project administration, J.L. and Z.C.; Resources, J.L. and Z.C.; Supervision, J.L. and Z.C.; Writing—original draft, Z.W.; Writing—review & editing, Z.W., J.L., Z.C., Q.Y. and X.X. All authors have read and agreed to the published version of the manuscript.

**Funding:** This work was funded by the National Key R&D Program of China (2019YFC0605203), the National Natural Science Foundation of China (Grant No. 41872096) and the Chinese National Non-Profit Institute Research Grant of CAGS (JYYWF201814).

**Institutional Review Board Statement:** Not applicable.

**Informed Consent Statement:** Not applicable.

**Data Availability Statement:** The data presented in this study are partially available on request from the corresponding author.

**Acknowledgments:** We are grateful to the staff of the Sichuan Geological Survey for their assistance during the field seasons.

**Conflicts of Interest:** The authors declare that they have no known competing financial interest or personal relationships that could have appeared to influence the work reported in this article.

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
