# Peer review of "Evolution and Li Mineralization of the No. 134 Pegmatite in the Jiajika Rare-Metal Deposit, Western Sichuan, China: Constrains from Critical Minerals"

_minerals, doi:10.3390/min12010045_

Round 1
Reviewer 1 Report
The paper is pleasant to read up to the results section, written in an acceptable English, well illustrated and referenced. The English grammar suddenly gets worse in the results section, but the authors should be able to correct that. The manuscript also suffers from some missing parts like Figure 7. Otherwise, the chemical data are of quality, and some original textural data is shown, like the mineral assemblage beryl-apatite showing uncommon alteration features. The crystallisation model is well argued.
I don't see why the section “3. Mineralogical and Petrographic Characteristic of the No.134 Pegmatite” is not in the Results section. Mineral descriptions are given in these two sections, so it is confusing to follow. The results could be restructured into a petrographic section where only hand sample description is given, and a mineralogical section where textures and chemistry are described together.
It would be good to see some EPMA points located on the BSE images. You could then give a few EMPA analysis in the paper and keep the full dataset as supplementary material. For example, we could then see the chemical difference between primary and secondary spodumene. In addition, this distinction between primary and secondary spodumene is not visible in the EPMA data table. If you calculate average compositions for all rim and all middle points, there is no clear difference in the analyses.
You state (lines 242-244) that “an obvious pattern of the chemical changes can be concluded as: from the core to the outer ring and/or the rim in the zoned crystals, femic components (FeO+MnO) decrease and Li2O content increase. Generally, the altered rim of spodumene contains about 0.1 wt% higher amount of Li2O than that of primary spodumene.” I don't see that in the EPMA results, and a 0.1 wt% Li2O difference can not be significative given the uncertainties in the determination of Li (which you note line 239). Consequently, there is no chemical justification for distinguishing primary versus secondary spodumene. This distinction is only textural, unless you have the chemical data to prove that it is also chemical.
Detailed corrections:
Lines 33-37: too long sentence, rewrite.
Lines 38-39: replace “resulted from the intruding of pegmatitic melt along the fracture” by “triggered by the intrusion of pegmatitic melt along the fractures”
Lines 39-40: replace “which could also indicate the process of evolution of other spodumene pegmatites” by “which could also be the dominant evolution process for other spodumene pegmatites”
Line 42: replace “clearly texturally zoning” by “clear textural zoning”
Line 45: replace “by minimize Li transfer to the country rocks through diffusion” by “by minimizing diffusional Li transfer to the country rocks”
Line 51: remove “in the area”
Line 51: remove “Generally”
Line 83: can you precise which metamorphic minerals? High-pressure? High-grade or low-grade?
Line 88: remove “afterwards”
Line 106: add an error to the 195.7 Ma age.
Line 110: replace “were evaluated to have exceeded” by “was evaluated to exceed”
Line 123: replace “is listed” by “are listed”
Line 132: remove “successive”
Line 136: replace “beryls are” by “beryl is”
Line 142: if these minerals have been studied elsewhere, please add a reference.
Line 148: replace “were short” by “makes short”
Line 149: replace “commonly shows” by “commonly with”
Line 151: replace “Muscovite are generally” by “Muscovite generally consists of”
Lines 159-161: reword the sentence: “Spodumene is commonly broken and shows zoning patterns. Cracks and fractures may be filled by cookeite and muscovite, and may contain mineral inclusions such as albite and cassiterite. It also occasionally displays symplectic rims (Fig. 4c).”
Line 162: replace “altered into” by “altered to”
Line 163: replace “Accessories including” by “Accessories include”
Line 167: replace “Spodumene are” by “Spodumene is”
Line 168: replace “commonly presented” by “common”
Line 178: replace “spodumene were short to elongate laths, mostly ranges within 0.1~0.2 mm” by “where spodumene forms short to elongate laths, mostly ranging from 0.1 to 0.2 mm”
Line 179: replace “ranges 2~4 cm” by “ranging from 2 to 4 cm”
Line 186: replace “oscillatory zoning core” by “oscillatory zoned core”
Figures 4 and 5: scale is a bit small
Lines 206-207: “The detection limits for most elements during this EPMA analysis are 40×10-6-200×10-6.” Please check detection limits, you mean 0.004 to 0.02 wt% oxides?
Line 209: delete “that there are”
Line 210: delete “present”
Line 210: delete “minerals including”
Line 215 and elsewhere: check that mineral name is given in singular (e.g. beryl and not beryls) and accord the verb. Here replace “They are absent” by “It is absent”. Plural may be used if you refer to grains, crystals etc... e.g. beryl crystals.
Line 218: replace “of them” by “of these minerals”
Line 220: “intersecting”
Line 221: “they may display”
From there to the end, the text suffers from many grammatical mistakes, please check.
Line 262: Fig. 7 is cited before Fig. 6.
Line 321: replace “Li-saturated Pegmatitic Melt” by “Li-saturated Pegmatite-forming Melt”. The term Pegmatite refers to a texture and cannot be applied to a chemistry.
Line 490: Fig. 9 should be Fig. 10?
Comments on figures, references and supplementary material:
References 26 and 27: please replace “Lichtervelde” by “Van Lichtervelde” (also for ref. 25) and move to the V letter.
Figure 2: I only have the geological section while a Geological map (a) is announced in the legend.
There is no Figure 7.
Figure 9: dots are too small, please change symbol color, shape and size. You should keep the same symbols between Figures 8 and 9.
Figure 11: what is the difference between the right and left figures?
Supplementary Table 3: remove the last 2 columns.
Supplementary Tables: EPMA data below detection limits should be declared “bdl” in tables
Author Response
Point 1: The manuscript also suffers from some missing parts like Figure 7. Otherwise, the chemical data are of quality, and some original textural data is shown, like the mineral assemblage beryl-apatite showing uncommon alteration features. The crystallisation model is well argued.
Response 1: Thank you for your comments. Figure 7 has been omitted and is now complete.
Point 2: I don't see why the section “3. Mineralogical and Petrographic Characteristic of the No.134 Pegmatite” is not in the Results section. Mineral descriptions are given in these two sections, so it is confusing to follow. The results could be restructured into a petrographic section where only hand sample description is given, and a mineralogical section where textures and chemistry are described together.
Response 2: The results section is restructured.
Point 3: It would be good to see some EPMA points located on the BSE images. You could then give a few EMPA analysis in the paper and keep the full dataset as supplementary material. For example, we could then see the chemical difference between primary and secondary spodumene. In addition, this distinction between primary and secondary spodumene is not visible in the EPMA data table. If you calculate average compositions for all rim and all middle points, there is no clear difference in the analyses.
Response 3: EPMA points and the corresponding analysis point number is now located on the BSE images, and the full dataset can be queried in the supplementary material.
Point 4: You state (lines 242-244) that “an obvious pattern of the chemical changes can be concluded as: from the core to the outer ring and/or the rim in the zoned crystals, femic components (FeO+MnO) decrease and Li2O content increase. Generally, the altered rim of spodumene contains about 0.1 wt% higher amount of Li2O than that of primary spodumene.” I don't see that in the EPMA results, and a 0.1 wt% Li2O difference can not be significative given the uncertainties in the determination of Li (which you note line 239). Consequently, there is no chemical justification for distinguishing primary versus secondary spodumene. This distinction is only textural, unless you have the chemical data to prove that it is also chemical.
Response 4: The results show that the chemical composition of spodumene in each subzone of zone Ⅱ vary irregularly, however, within individual zoned spodumene crystals, an obvious pattern of the chemical changes can be concluded as: from crystal core to the outer ring and/or the rim in the zoned crystals, femic components (FeO+MnO) decrease and Li2O content increase. Generally, the altered rim of a spodumene crystal contains about 0.1 wt% higher amount of Li2O than its core. However, this law is not applicable to the comparison between different crystals, there is no chemical justification for distinguishing rim versus core chemistry of spodumene. Other components including Fe, and lesser Sr, Na, and Mn, occur as small amounts of impurities in chemical compositions of spodumene (supplementary Table 1).
Point 5: Figures 4 and 5: scale is a bit small
Response 5: Modified.
Point 6: “The detection limits for most elements during this EPMA analysis are 40×10-6-200×10-6.” Please check detection limits, you mean 0.004 to 0.02 wt% oxides?
Response 6: We confirm that the detection limit is the detection limit of elements, and the value range is 200-40ppm.
Point 7: From the Results section to the end, the text suffers from many grammatical mistakes, please check.
Response 7: Grammatical mistakes were carefully revised.
Point 8: I only have the geological section while a Geological map (a) is announced in the legend
Response 8:The legend is revised.
Point 9: Figure 9: dots are too small, please change symbol color, shape and size. You should keep the same symbols between Figures 8 and 9.
Response 9: Have changed the symbol color, shape and size in Figures 8 and 9, and keep them the same.
Point 10: Figure 11: what is the difference between the right and left figures?
Response 10: Figure 11: the missing part is now complete.
Point 11: Supplementary Table 3: remove the last 2 columns;EPMA data below detection limits should be declared “bdl” in tables
Response 11: All revised.

Reviewer 2 Report
In revision of the manuscript entitled “Evolution and Li Mineralization of the No.134 Pegmatite in the Jiajika Rare Metal Deposit, Western Sichuan, China: Constrains From Critical Minerals” by Wang et all.
The Manuscript deals with pegmatite evolution study No.134 containing lithium at the Jiajika Deposit, West Sichuan, China.
The author carefully describes textural relations and mineral paragenesis. The pegmatite zoning has been well reported. Mineral chemistry was done by EMPA. Mineral evolution is well described in Table 1. Results and interpretations are well supported by a complete literature review.
The manuscript is very interesting, well written and organized. The figures are representative and of good quality. It is an important addition in the literature of Li mineralized pegmatites,
I highly recommend the manuscript for publication in Minerals.
Author Response
Dear reviewer,
Thank you very much for your comments!
Kind regards,
Dr. Wang
BGRIMM Technology Group
Round 2
Reviewer 1 Report
Line 206-207: “The yellow dot indicates the location of the analysis point, and the corresponding name of the analysis point can be found in supplementary table 2.” change to “Location of analytical points is indicated by yellow dots; the corresponding names of the analytical points can be found in supplementary table 2.”
Lines 209-219: this part should go into a methodology section before the Results.
Line 219: add unit for detection limits.